# Predicting dietary management intention of patients with chronic kidney disease using protection motivation theory

**Huijie Li**[1], **Yueyi Deng**[1], **Yitong Huang**[2]*, **Holly Blake**[3,4]

**1** Department of Nephrology, Longhua Hospital, Shanghai University of Traditional Chinese Medicine, Shanghai, China, **2** School of Media and Communication, Shanghai Jiaotong University, Shanghai, China, **3** School of Health Sciences, University of Nottingham, Nottingham, United Kingdom, **4** NIHR Nottingham Biomedical Research Centre, Nottingham, United Kingdom

\* huang.yitong@foxmail.com

## Abstract

### Background

Psychological determinants underlying the dietary management intention (DMI) of Chinese patients with chronic kidney disease (CKD) are not well understood. This hinders the development of theory-informed dietary interventions targeting this population. The aim of this study was to identify factors influencing DMI of Chinese patients with CKD through the lens of Protection Motivation Theory (PMT).

### Methods

500 patients with CKD from a nephrology ward of a large teaching hospital in China completed a survey including measures of PMT constructs (i.e., perceived vulnerability, perceived severity, intrinsic and extrinsic rewards, self-efficacy, response efficacy, and response cost) using validated scales adapted from previous studies. Data were analyzed using confirmatory factor analysis and multiple linear regression.

### Results

Three PMT constructs, namely perceived severity [B=0.198, $P < 0.001$], response efficacy [B=0.331, $P < 0.001$], and self-efficacy [B=0.325, $P < 0.001$], two demographic variables, namely single status [B=-0.180, $P = 0.028$] and education level [B=0.080, $P = 0.007$], and a disease-related variable, namely CKD stage [B=.056, $P = 0.001$], predicted 39.3% of the variance of the CKD DMI. No significant effect on CKD DMI was observed for other predictor variables ($P > 0.05$).

### Conclusions

Applying the PMT, significant predictors of DMI in Chinese patients with CKD were identified, which should be targeted in behavior change initiatives aimed at promoting dietary management.

**Data availability statement:** All relevant data are within the manuscript and its Supporting Information files.

**Funding:** This research was funded by Shanghai Municipal Health Commission through its Shanghai Municipal Key Clinical Specialty Program (grant number SHSLCZDZK04201, awarded to Y.D.). The funder website can be accessed at https://wsjkw.sh.gov.cn/. The funder had no role in study design, data collection and analysis, decision to publish, or preparation of the manuscript.

**Competing interests:** The authors have declared that no competing interests exist.

**Abbreviations:** AMOS, Analysis of Moment Structure;  AVE, Average Variance Extracted; CFA, Confirmatory Factor Analysis; CFI, Comparative Fit Index; CKD, Chronic Kidney Disease; CMID/DF, Chi-square to Degree of Freedom Ratio; CR, Composite Reliability; DASH, Dietary Approaches to Stop Hypertension; DMI, Dietary Management Intention; eGFR, Estimated Glomerular Filtration Rate; ER, Extrinsic Reward; IFI, Incremental Fit Index; IR, Intrinsic Reward; PMT, Protection Motivation Theory; PS, Perceived Severity; PV, Perceived Vulnerability; RC, Response Cost; RE, Response Efficacy; RMSEA, Root Mean Square Error of Approximation; SE, Self-efficacy; TLI, Tucker-Lewis Index.

## Introduction

Chronic kidney disease (CKD) is a pathological condition in which a gradual loss of kidney function occurs. It is clinically classified into five stages based on estimated glomerular filtration rate (eGFR) [1]. As patients in early stages of CKD might have few signs or symptoms, they may feel reluctant to seek and follow medical advice on managing the condition. Lack of early interventions can lead to progression of CKD to stages characterized by irreversible nephron loss. End-stage renal disease is fatal without dialysis or kidney transplant, both of which impose significant burdens on patients and their families, as well as the healthcare systems and society at large, diverting resources away from other medical priorities [2,3]. The issue is exacerbated by the escalating prevalence of CKD [1]. Globally, 13.4% of the population are diagnosed with CKD, and millions die each year because of a lack of access to affordable treatment [4]. There are approximately 132 million cases of CKD in China, accounting for around one-fifth of the global CKD burden; this figure will increase disproportionally in the coming years due to China's aging population [5]. The prevalence of CKD in China is estimated to reach 165 million in 2025, costing the economy $198 billion per year [6]. It is therefore imperative to engage patients with CKD in effective interventions and management of their condition from the earliest stage possible.

Modifiable behavioral risk factors associated with CKD development and progression, include (but are not limited to) unhealthy eating, smoking, and use of nephrotoxic substances. As such, patient self-management and behavior change is critical for condition management and has been recognized as an integral component of most CKD treatments [7] with proven effectiveness for improving clinical outcomes in both pre-dialysis and dialysis patients with CKD [8,9].

Among all the self-management components, diet is of particular importance to preventing CKD development and progression. Four systematic reviews with meta-analysis of prospective cohorts and cross-sectional studies have reported an inverse association between adherence to healthy dietary patterns and the development and progression of CKD [10–13]. Despite the different terminologies (e.g., grains-vegetables diet, Mediterranean diet, and the Dietary Approaches to Stop Hypertension (DASH) diet) employed to denote the concept of healthy dietary patterns in these studies, they all converge on several shared characteristics, including an emphasis on whole grains, fruits and vegetables, limited intake of red meat, processed food and sweetened beverages, and a balanced incorporation of plant-based fats and lean protein [10,11,13]. As for clinical populations diagnosed with CKD (or conditions associated with high risk for developing CKD such as diabetes and hypertension), dietary management is a more complicated issue and needs to balance different nutritional needs. Although there are controversies around the benefits of nutritional supplements [14,15], there is strong and consistent evidence for the nephroprotective effects of dietary modifications with an emphasis on salt and protein restriction in patients with CKD [16]. A high-protein diet, defined as intake of > 1.2 g protein per kg of body weight per day (g/kg/day) can accelerate renal impairment and aggravate uremic symptoms in patients with CKD [17]. Therefore, a low-protein diet (0.6 to 0.8 g/kg/day) with half of the protein from high quality sources such as eggs, fish and poultry is recommended to most patients with CKD [18], though the optimal amount should be tailored to individual patients based on nutritional need assessment by a clinician or dietitian [19]. A salt-reduction regime can also benefit patients with CKD via controlling fluid retention, lowering blood pressure, and reducing risks for cardiovascular risks, according to conclusions from systematic reviews [16].

Although the need for dietary modifications in CKD management is well-established, a notable gap persists in the provision of effective dietary interventions to Chinese patients with CKD [20,21]. While Chinese patients with CKD have been routinely advised by nephrology physicians and nurses to engage in dietary monitoring and management, compliance with dietary regimes is unsatisfactory [16,22]. There are myriad reasons, not least the fact that renal diet is arguably

the most restrictive and difficult one to follow [23], but also that current dietary recommendations tend to be based on a medical model which fails to account for the science of behavior change [20,24]. Developing and implementing successful behavior change interventions requires a thorough understanding of the psychological mechanisms underlying the target behavior [25]. When it comes to dietary management of patients with CKD, there is a lack of theoretically driven research on the psychological determinants of dietary management intention (DMI), beyond demographic, socioeconomic and disease-related characteristics [20,26,27].

To address this knowledge deficit and to shed light on the intricacies of dietary management behavior of Chinese patients with CKD, we adopted the Protection Motivation Theory (PMT) as an instrumental framework. The PMT is a psychological model that explains how people are motivated to react in a self-protective way towards events with potential harms [28]. PMT has been widely applied in research focused on enhancing the understanding of protective motivations in diverse health domains [29,30]. The theory posits that behavioral intention to adopt protective behavior is determined by threat appraisals and coping appraisals. Threat appraisals encompass perceived severity of the negative consequences of the threatening event, perception of their own vulnerability to the threatening event, intrinsic and extrinsic rewards of the maladaptive behavior. In the context of this study, perceived severity refers to a patient's perceived adverse impacts of renal decline on both themselves and their families. Perceived vulnerability refers to the patients' perceived likelihood or probability that renal decline and complication affect them personally under the circumstance of an uncontrolled diet. Coping appraisal, on the other hand, encompasses self-efficacy, response efficacy, and response cost [31]. Response efficacy refers to the extent to which patients with CKD believes that good dietary management can prevent their condition from deteriorating, whereas self-efficacy concerns their belief about their own ability to adhere to the recommended diet proficiently. The response costs are the costs associated with adhering to a renal diet. The protection motivation (i.e., the DMI) will be high if perceived severity, vulnerability, self-efficacy, and response efficacy are high, and rewards for uncontrolled diet and costs for restricted diets are low.

Even though the PMT has been successfully employed to predict behavioral intention in various health domains (e.g., the COVID-19 vaccination [32], self-management in diabetes [33] and cardiovascular disease [34]), no prior studies have explored how PMT can be used to understand DMI in patients with CKD. With the rising prevalence of CKD in China, developing a theory-informed understanding of factors that influence dietary management in Chinese patients with CKD is critical. Therefore, the aim of the study is to examine factors influencing DMI of Chinese patients with CKD through the lens of Protection Motivation Theory (PMT). Based on PMT, we hypothesized that intention to engage in CKD dietary management would be:

H1. positively predicted by perceived vulnerability of CKD progression.

H2. positively predicted by perceived severity of CKD progression.

H3. negatively predicted by intrinsic rewards for uncontrolled diet.

H4. negatively predicted by extrinsic rewards for uncontrolled diet.

H5. positively predicted by response efficacy, or the belief that dietary management would work to delay or prevent CKD progression.

H6. positively predicted by self-efficacy, or the belief in one's ability in engaging in CKD dietary management.

H7. negatively predicted by response cost, or the perceived barriers to maintaining a CKD-friendly diet.

## Materials and methods

### Procedure

We adhered to the STROBE reporting guidelines for cross-sectional studies [35] (see checklist in S1 Appendix). Data were collected as part of a larger project on self-management of patients with CKD, which received approval from the Research Ethics Committee of Longhua Hospital (ref: 2021LCSY107). Participants were recruited from the nephrology ward at the participating hospital. Patients were eligible for the study if they were over 18 years old, and had a diagnosis of CKD at any stage. The diagnosis was based on the criteria specified in the Clinical Practice Guideline for the Management of Glomerular Diseases [36], which include the presence of either a GFR less than 60 ml/min/1.73m² for more than three months, or signs of kidney damage (e.g., proteinuria, structural abnormalities observed in imaging) persistent for the same duration. Patients who had dementia or any documented mental or cognitive disorder were excluded from the study. Data were collected in-person between October 2021 and November 2022, with direct approach on the ward by the researcher. Of the 1328 potentially eligible patients with CKD approached, 500 (37.7%) patients consented to participate and completed the survey. Reasons for non-participation included being critically ill and unconscious, being prevented from taking part by families, concerns about privacy, cognitive impairment and lack of interest. The researcher explained the purpose and requirements of the study and was available to answer participant's questions about the research. Participants were informed that they were able to withdraw from the study with no impact on their clinical care. Consenting participants completed the survey independently, unless they required support from the researcher (e.g., if they had vision problems or difficulties reading or understanding). They were required to complete the survey on a single occasion, and it took approximately 30 minutes.

### Inclusivity in global research

Additional information regarding the ethical, cultural, and scientific considerations specific to inclusivity in global research is included as S2 Appendix.

### Variables and measurement

The survey collected demographic and CKD-related information including age, gender, height and weight, marital status, education level, employment status and occupation, monthly income, CKD stage and duration since first diagnosis. We adapted scales used in previous studies [37–39] and mapped them onto the following PMT constructs respectively. All items were rated on a 5-point Likert scale, where 1 represented complete disagreement, and 5 represented complete agreement. Full text for the PMT items (in English) is available in S3 Appendix.

Perceived vulnerability: 3 items measuring the dimension of "perceived vulnerability" from the Healthy Eating Cognitive Beliefs Questionnaire for CKD [37] were used (example item: "Loss of renal function is likely to happen to me if I do not follow a renal diet recommended by the clinician").

Perceived Severity: 6 items measuring the dimension of "perceived severity" from the original questionnaire [37] were used (example statements: "Renal decline will have a negative impact on [the everyday life and work of my family and caregivers/my own everyday life and work]"; "Renal decline will increase the chance of requiring frequent hospitalizations and hospital visits").

Intrinsic rewards (of the maladaptive behavior): we referred to wording used in other PMT-informed survey studies [38,39] and derived 5 items to assess intrinsic rewards for an unrestricted diet (example items: "I think an unrestricted diet makes me [feel more relaxed/enjoy life more/feel better about myself]").

Extrinsic rewards (of the maladaptive behavior): we referred to wording used in other PMT-informed survey studies [38,39] and derived 4 items to assess extrinsic rewards for an unrestricted diet (example item: "My family think it is easier to prepare food if I maintain an unrestricted diet").

Response efficacy: 5 items measuring the dimension of "positive outcome expectancy" of a healthy diet in the aforementioned validated questionnaire [37] were adopted (example item: "Following a healthy diet can slow the impairment of kidney function").

Self-efficacy: 5 items measuring the dimension of "self-efficacy" from the original questionnaire [37] were adopted (example statement: "I have the knowledge and skills required for getting the right type and amount of food suitable for my condition").

Response cost: 5 items measuring the dimension of "negative outcome expectancy" from the original questionnaire [37] were used (example item: "Following a renal diet hinders my social life").

Dietary Management Intention: 5 items from a validated questionnaire [37] were used (example item: "I intend to follow a healthy diet after hospital discharge").

## Statistical analysis

Descriptive statistics and regression analysis were performed in SPSS version 22.0 (IBM Corp., Armonk, NY, USA). The measurement model for the PMT constructs was validated by performing confirmatory factor analysis (CFA) [40] in the Analysis of Moment Structures (AMOS) software version 24 (IBM Corp., Armonk, NY, USA). The model was estimated based on maximum likelihood method and assessed using fit indices such as the Chi-square to degrees of freedom ratio (CMIN/DF), comparative fit index (CFI), Tucker-Lewis index (TLI), incremental fit index (IFI), and the root mean square error of approximation (RMSEA) [41]. The measurement model was refined based on modification indices and standardized factor loading estimates following a conservative strategy [40]. To assess the predictive power of all PMT constructs on DMI while controlling for potential influences of demographic and disease-related characteristics, we chose regression analysis over structural equation modelling. This decision was based on the ability of regression analysis to include and report both significant and non-significant predictors. Diagnostic tests were conducted to verify that all assumptions required for regression analysis were satisfied.

## Results

### Participant characteristics

Table 1 shows the sociodemographic and health characteristics of the participants. Participants' age ranged from 18 to 92 years old. Most of the participants were married (87.8%), educated to high school level or below (61.6%) and had comorbidities (88%) alongside their CKD. Over two-thirds of the participants were not in employment.

### Confirmatory factor analysis

Based on CFA results, we removed three items with factor loadings lower than 0.5, namely: intrinsic reward (IR)-1 (Factor loading = 0.174), intrinsic reward (IR)-5 (Factor loading = 0.132), and response cost (RC)-1 (Factor loading = 0.485) from the measurement model. The resulting measurement model showed acceptable fit indices (CMIN/DF = 2.364, CFI = 0.943, TLI = 0.936, IFI = 0.943, RMSEA = 0.052) according to established cutoff values

**Table 1. Profile of participants.**

| Characteristics | N (%) or M ± SD |
| --- | --- |
| **Gender** | |
| Male | 275 (55.0) |
| Female | 225 (45.0) |
| **Age** | 57.02 ± 14.41 |
| **Height (centimeters)** | 165.42 ± 13.78 |
| **Weight (kilograms)** | 66.00 ± 13.87 |
| **Duration of CKD (months)** | 94.41 ± 92.14 |
| **Stage of CKD** | |
| 1 | 97 (19.4) |
| 2 | 74 (14.8) |
| 3 | 96 (19.2) |
| 4 | 75 (15.0) |
| 5 | 158 (31.6) |
| **Marital status** | |
| Married | 439 (87.8) |
| Unmarried | 27 (5.4) |
| Divorced | 11 (2.2) |
| Widowed | 23 (4.6) |
| **Educational level** | |
| Below Primary school | 6 (1.2) |
| Primary school | 26 (5.2) |
| Middle school | 114 (22.8) |
| High school/Technical secondary school | 162 (32.4) |
| College/ University | 179 (35.8) |
| Postgraduate | 13 (2.6) |
| **Employment status** | |
| Employed | 145 (29.0) |
| Retired | 284 (56.8) |
| Laid off or unemployed | 50 (10.0) |
| Student | 2 (0.4) |
| Other | 19 (3.8) |
| **Occupation or pre-retirement occupation** | |
| Professional and technical personnel | 83 (16.6) |
| Service personnel | 45 (9.0) |
| Freelancer | 31 (6.2) |
| Worker | 92 (18.4) |
| Agriculture, animal husbandry, fishing and mining | 7 (1.4) |
| Public institution | 54 (10.8) |
| Company employee | 120 (24.0) |
| Other | 68 (13.6) |
| **Care provided by** | |
| Self-care | 343 (68.6) |
| Spouse | 119 (23.8) |
| Child(ren) | 27 (5.4) |
| Other people | 11 (2.2) |
| **Monthly income (RMB/person)** | |
| <2000 | 37 (7.4) |

*(Continued)*

**Table 1.** (Continued)

| Characteristics | N (%) or M ± SD |
|---|---|
| [2000, 4000) | 102 (20.4) |
| [4000, 6000) | 139 (27.8) |
| [6000, 8000) | 68 (12.6) |
| [8000, 10000) | 52 (10.4) |
| >= 10000 | 102 (20.4) |
| **Complication and comorbidities (multiple choice)** | |
| Diabetes | 156 (31.2) |
| Hypertension | 156 (31.2) |
| Cardiovascular disease | 89 (17.8) |
| None | 86 (17.2) |
| Other | 38 (7.6) |

Note. Categorical variables are presented as counts and percentages, whereas continuous variables are presented as M ± SD; N (%) represents the count (N) and its corresponding percentage of the total; M represents mean and SD represents standard deviation. An approximate exchange rate of 7.1 RMB to 1 USD has been applied, acknowledging potential fluctuations in exchange rates.

(i.e., CMIN/DF < 3, CFI > 0.9, TLI > 0.9, IFI > 0.9, RMSEA < 0.08 [40,41]). Cronbach's alpha for PMT constructs ranged from 0.807 to 0.938, exceeding the recommended level of 0.60, demonstrating internal consistency reliability. Detailed descriptive statistics, factor loadings, and Cronbach's α for the modified variables are presented in Table 2.

As shown in Table 3, composite reliability (CR) for each construct ranged from 0.821 to 0.938, exceeding the cutoff value of 0.5, which suggested that all latent variables had good convergent validity. The average variance extracted (AVE) for each latent variable ranged from 0.540 to 0.753, exceeding the recommended cutoff value of 0.5 [42]. This, along with the fact that the square root of the AVE value was greater than the correlation coefficients between the corresponding latent variables and other latent variables, suggested good discriminant validity of these variables [42].

## Hypothesis testing

As shown in Table 4, perceived severity, response efficacy, self-efficacy, single status, education level, and CKD stage were significant predictors in the regression model, which explained 39.3% of the variance in DMI. Specifically, the greater the perceived severity, response efficacy, and self-efficacy, the greater the intention to engage in dietary management. Moreover, the model controlled for age, single status, employment status, education level, gender, income and CKD stage. Results indicated that only three control variables (i.e., education level, single status, and CKD stage) were significant predictors of DMI. Specifically, high education level and advanced CKD stage positively predicted DMI, whereas being single negatively predicted DMI. All assumptions for multiple linear regression (e.g., linearity, no concerning multicollinearity, independence, normality and equal variance of residuals etc.) were satisfied. Table 5 summarizes the hypothesis testing results.

## Discussion

### Main findings

The primary aim of this study was to examine factors influencing DMI among Chinese patients with CKD through the lens of PMT, shedding light on the nuanced process of

**Table 2. Descriptive statistics, factor loadings and Cronbach's α for the PMT scales.**

| Measures | Items | M ± SD | M ± SD | Factor loading | Cronbach's α |
|---|---|---|---|---|---|
| **Perceived vulnerability** | PV 1 | 4.612 ± 0.845 | 4.622 ± 0.716 | 0.771 | 0.835 |
| | PV 2 | 4.590 ± 0.871 | | 0.755 | |
| | PV 3 | 4.640 ± 0.800 | | 0.871 | |
| **Perceived severity** | PS 1 | 4.697 ± 0.734 | 4.690 ± 0.620 | 0.891 | 0.898 |
| | PS 2 | 4.669 ± 0.758 | | 0.842 | |
| | PS 3 | 4.752 ± 0.675 | | 0.942 | |
| | PS 4 | 4.480 ± 1.008 | | 0.523 | |
| | PS 5 | 4.626 ± 0.914 | | 0.622 | |
| | PS 6 | 4.624 ± 0.929 | | 0.615 | |
| **Intrinsic rewards** | IR 2 | 2.018 ± 1.414 | 2.209 ± 1.257 | 0.616 | 0.857 |
| | IR 3 | 2.258 ± 1.421 | | 0.966 | |
| | IR 4 | 2.352 ± 1.44 | | 0.899 | |
| **Extrinsic rewards** | ER 1 | 2.398 ± 1.508 | 2.001 ± 0.906 | 0.542 | 0.807 |
| | ER 2 | 1.504 ± 1.105 | | 0.72 | |
| | ER 3 | 1.594 ± 1.158 | | 0.808 | |
| | ER 4 | 1.739 ± 1.261 | | 0.833 | |
| **Response efficacy** | RE 1 | 4.570 ± 0.786 | 4.602 ± 0.657 | 0.839 | 0.905 |
| | RE 2 | 4.640 ± 0.785 | | 0.782 | |
| | RE 3 | 4.566 ± 0.799 | | 0.709 | |
| | RE 4 | 4.589 ± 0.769 | | 0.827 | |
| | RE 5 | 4.644 ± 0.717 | | 0.912 | |
| **Self-efficacy** | SE 1 | 3.912 ± 1.061 | 4.028 ± 0.859 | 0.72 | 0.903 |
| | SE 2 | 3.655 ± 1.141 | | 0.647 | |
| | SE 3 | 3.919 ± 1.075 | | 0.943 | |
| | SE 4 | 3.922 ± 1.080 | | 0.916 | |
| | SE 5 | 3.828 ± 1.156 | | 0.773 | |
| **Response cost** | RC 2 | 3.529 ± 1.323 | 3.207 ± 1.097 | 0.643 | 0.837 |
| | RC 3 | 3.093 ± 1.345 | | 0.902 | |
| | RC 4 | 2.951 ± 1.362 | | 0.875 | |
| | RC 5 | 3.254 ± 1.327 | | 0.586 | |
| **Dietary management intention** | DMI 1 | 4.354 ± 0.963 | 4.461 ± 0.745 | 0.834 | 0.938 |
| | DMI 2 | 4.476 ± 0.797 | | 0.943 | |
| | DMI 3 | 4.434 ± 0.829 | | 0.957 | |
| | DMI 4 | 4.572 ± 0.75 | | 0.831 | |
| | DMI 5 | 4.569 ± 0.758 | | 0.759 | |

intention formation surrounding CKD dietary management behaviors. The PMT is broadly used as a social cognitive theory in predicting health behaviors [29,43,44] and in guiding research to devise intervention programs for purposeful behavior change [33,45]. However, we found no scholarly articles published in English drawing on the PMT to understand or intervene with CKD-related dietary management in a Chinese sample. To our knowledge, our study is the first to measure and model psychological constructs underlying the DMI of Chinese patients with CKD based on the PMT. We adapted the Healthy Eating Cognitive Beliefs Questionnaire for Chronic Kidney Disease, which was previously developed and validated for assessing psychological variables that overlapped with PMT constructs [37]. The validity and reliability of the instrument in assessing PMT constructs was affirmed through the rigorous CFA procedure. In addition, our results revealed a positive association between perceived

Table 3. Bivariate correlations and validity metrics of main variables.

| | CR | AVE | PV | PS | IR | ER | RE | SE | RC | DMI |
|---|---|---|---|---|---|---|---|---|---|---|
| PV | 0.842 | 0.641 | 0.726 | | | | | | | |
| PS | 0.884 | 0.571 | 0.274** | 0.756 | | | | | | |
| IR | 0.875 | 0.707 | -0.145** | -0.054 | 0.841 | | | | | |
| ER | 0.821 | 0.540 | -0.294** | -0.143** | 0.285** | 0.735 | | | | |
| RE | 0.909 | 0.667 | 0.467** | 0.422** | -0.130** | -0.133** | 0.773 | | | |
| SE | 0.902 | 0.653 | 0.246** | 0.253** | -0.057 | -0.06 | 0.445** | 0.808 | | |
| RC | 0.844 | 0.584 | -0.059 | 0.048 | 0.243** | 0.105* | 0.025 | -0.055 | 0.764 | |
| DMI | 0.938 | 0.753 | 0.276** | 0.351** | -0.054 | -0.076 | 0.478** | 0.534** | -0.019 | 0.868 |

Note. CR = composite reliability, AVE = average variance extracted; values in bold are Square root of AVE;

* $p < 0.05$;

** $p < 0.001$. Note: The bold values in diagonal represent the sqrt (AVE) values.

Table 4. Standardized coefficients of pathways between PMT constructs for CKD individuals.

| Predictor | B | SE | t | P value |
|---|---|---|---|---|
| Constant | 0.445 | 0.366 | 1.216 | 0.225 |
| Perceived Vulnerability | -0.060 | 0.038 | -1.552 | 0.121 |
| Perceived Severity | 0.198 | 0.047 | 4.200 | 0.000*** |
| Intrinsic Rewards | 0.010 | 0.023 | 0.436 | 0.663 |
| Extrinsic Rewards | -0.002 | 0.031 | -0.071 | 0.943 |
| Response Efficacy | 0.331 | 0.052 | 6.325 | 0.000*** |
| Self-Efficacy | 0.325 | 0.034 | 9.448 | 0.000*** |
| Response Cost | -0.011 | 0.025 | -0.430 | 0.667 |
| Age | 0.003 | 0.002 | 1.512 | 0.131 |
| Single Status | -0.180 | 0.082 | -2.207 | 0.028* |
| Employment | 0.003 | 0.074 | 0.037 | 0.970 |
| Education Level | 0.080 | 0.030 | 2.715 | 0.007* |
| Gender | 0.041 | 0.054 | 0.754 | 0.451 |
| CKD stage | 0.056 | 0.018 | 3.228 | 0.001** |
| Income | -0.021 | 0.019 | -1.135 | 0.257 |

Note.

* $P < 0.05$;

** $P < 0.01$;

*** $P < 0.001$. $R^2 = 39.3\%$. Single status (0 = not single, 1 = single (including unmarried, divorced or widowed)); Employment (0 = retired or unemployed, 1 = employed); Gender (0 = male; 1 = female); education level (1 = Below primary level, 2 = Primary level, 3 = junior secondary level, 4 = high school or trade school level, 5 = College or university level, 6 = Masters or PhD level); CKD stage(1 = Stage 1, 2 = Stage 2, 3 = Stage 3, 4 = Stage 4, 5 = Stage 5); income (1 = less than 2000 monthly, 2 = 2000 to less than 4000 monthly, 3 = 4000 to less than 6000 monthly, 4 = 6000 to less than 8000 monthly, 5 = 8000 to less than 10000 monthly, 6 = 10000 and above monthly).

severity and DMI, lending support to H2. This resonated with findings from prior qualitative studies that identified risk perception as a main facilitator to adherence to dietary and fluid restrictions among patients with CKD [46–48], suggesting that a heightened sense of the severity of CKD progression indeed motivated patients to prioritize and adopt the protective action (i.e., CKD dietary adherence). Additionally, our study highlighted response efficacy as another important predictor of DMI, lending support to H5. This was consistent with previous research, where the same construct, variably termed as "perceived benefits" or "positive outcome expectancies" under the framing of other theoretical models (e.g., Health Belief

**Table 5. Summary of hypothesis testing results.**

| Hypotheses | Relationship | Result |
|---|---|---|
| H1 | Perceived Vulnerability → DMI | Rejected |
| H2 | Perceived Severity → DMI | Supported |
| H3 | Intrinsic Rewards → DMI | Rejected |
| H4 | Extrinsic Rewards → DMI | Rejected |
| H5 | Response Efficacy → DMI | Supported |
| H6 | Self-Efficacy → DMI | Supported |
| H7 | Response Cost → DMI | Rejected |

Model [49] or the Health Action Process Approach model [50]), was found to consistently predict adherence to salt-restricted diet in patients with CKD [51,52]. Further, our findings underscored the pivotal roles of self-efficacy (i.e., the confidence in one's ability to engage in effective dietary management) in shaping CKD DMI, which supported H6 and aligned with prior research that consistently demonstrated self-efficacy, or perceived behavioral control, as a strong predictor of self-management behavior, including but not limited to dietary management in patients with CKD [53–57].

Contrary to our hypotheses, we found that perceived vulnerability of CKD progression, intrinsic and extrinsic rewards for uncontrolled diet, and response cost, had negligible influence on DMI, which rejected H1, H3, H4 and H7. This indicates that while these factors might hold theoretical importance, their impact on patients' actual intention towards dietary management are perhaps subject to the broader socio-cultural contexts within which these patients navigate their dietary choices and lifestyle behaviors.

## Practical implications

This study has enriched our understanding of how threat and coping appraisals collectively shape CKD-related health decision-making and behavior change. Several practical implications can be drawn. First, with regards to perceived severity and response efficacy, our findings suggest that interventions should emphasize the health risks associated with inadequate dietary adherence and benefits associated with effective dietary management, thereby enhancing patients' awareness of the significant impact that their dietary choices can have on their health outcomes.

Second, our study has highlighted self-efficacy as a key determinant of engagement in CKD-related dietary management. Self-efficacy is an important concept in health self-management [58] and is known to be a predictor of adherence to dietary and self-care behavior in chronic conditions (e.g., end-stage renal disease [59]; diabetes [60,61]; hypertension [62]). However, very few studies to date have incorporated self-efficacy building within dietary interventions for patients with CKD [63–65]. Therefore, interventions can potentially lead to more sustained and successful dietary adherence by placing more emphasis on enhancing food literacy and beliefs about capabilities. This should be considered in light of the complexity of CKD-related dietary management and in relation to the traditional Chinese food culture. The dietary adjustment required for CKD management depends on many factors including their comorbidities [66], which renders most patients' food literacy inadequate and in turn hinders their self-efficacy [20]. Moreover, most CKD dietary programs have been developed for Western populations and their suitability for East Asian populations is insufficiently studied [23]. It is worth noting that historically, non-communicable chronic diseases such as obesity, type 2 diabetes, heart disease, and cancer were rare among non-Western populations who retained traditional diets and lifestyles; as these populations

transitioned toward industrialized diets and lifestyles, the prevalence of these chronic diseases surged [67]. Now that the study reveals many Chinese patients with CKD find it difficult to understand and adhere to the complicated CKD diet parameters that are primarily derived from research conducted in Western countries [68], there is a need to revisit, examine and identify the health-promoting elements inherent in traditional Chinese diet and develop more user-friendly dietary guidelines for Chinese patients with CKD.

Thirdly, our study has found a positive association between DMI and education level, which is in line with previous research [69,70]. Additionally, our study has identified single status (including unmarried, divorced and widowed) as a significant risk factor for low DMI. Perhaps patients who are single might encounter unique challenges in adhering to dietary management due to potential lack of social support and accountability [71]. Therefore, healthcare practitioners should pay special attention to subgroups of patients who are single and have lower levels of education, providing tailored support and strategies that address their specific circumstances and barriers.

Finally, the positive association between DMI and CKD stage reflects an increased commitment to dietary management as the disease progresses. However, this is likely a reactive response to worsening health, rather than the result of early preventive education. This finding signals a need for proactive interventional programs promoting dietary management in the earlier stages of CKD, as relying on heightened intention in advanced stages could mean missed opportunities to slow disease progression. Clinicians and dietitians should pay more efforts to fostering DMI in patients in early CKD stages, which may help delay progression and instill long-term compliance habits.

## Study strengths and limitations

The key strengths of our study are its grounding in behavioral theory, a diverse participant sample and the methodological rigor. Firstly, the application of the PMT framework provided a robust theoretical foundation, enhancing our understanding of the complex interplay between cognitive appraisals that shaped CKD DMI. Secondly, the study was conducted in a hospital nationally renowned for chronic disease treatment and care, which allowed for recruitment of a sample that was broadly representative of patients with CKD from different regions in Southern China and with diverse socioeconomic backgrounds. Furthermore, the use of validated measures and rigorous statistical analyses ensured the validity of our data, reinforcing the credibility of our findings.

Several limitations should be noted in interpreting and using insights drawn from our study. The first limitation of the study was the cross-sectional design which restricted our ability to establish causality between the identified predictors and DMI. Second, in terms of data source, our study relied on self-reported survey data, which might be susceptible to recall bias and social desirability bias. The self-selection sample also limits the generalizability of our results, as individuals who rejected to fill in the survey might possess distinct psychological or motivational characteristics compared to those who responded to the survey request. Finally, the recruitment of only patients restricted the generalizability of our findings to outpatient populations. However, this choice was justified by the need to accommodate the literacy levels of our respondents, ensuring they had sufficient time to read and understand the survey questions and seek clarification from the researcher if needed, which would have been difficult in outpatient settings which are often crowded and hectic in China. Moreover, it is quite common for CKD patients to alternate between inpatient and outpatient care, especially given that the hospital where the study was conducted was a Chinese medicine hospital specialized in chronic disease management.

## Implications for future research

Future studies should employ longitudinal designs to explore how the DMI of patients with CKD change over time and establish causal relationships between the identified predictors and dietary adherence. Qualitative research could also be conducted to offer insights into the lived experiences of patients with CKD, providing a deeper understanding of the barriers and enablers of adherence to CKD diets, and any contextual factors that influence their dietary decisions. Interventional studies targeting the identified determinants may be conducted to accumulate empirical evidence regarding effective theory-informed strategies for promoting dietary management adherence among Chinese patients with CKD. Based on our findings, theoretically-informed interventions may be developed and evaluated that (a) focus on the provision of education about the severity of CKD at different stages and associated complications, (b) present the benefits of adhering to CKD diets, (c) provide dietary advice for patients with CKD that is culturally relevant (i.e., accounting for the traditional Chinese diet), and (d) include strategies to build patients' confidence (or "self-efficacy") in adherence to dietary regimes for CKD. Furthermore, interventional research targeting specific groups may be warranted (e.g., patients who do not have support from a significant other, and who have low levels of education). Research along these lines can collectively contribute to the ongoing efforts to enhance the health outcomes and wellbeing of patients with CKD through dietary interventions and support programs.

## Conclusions

In summary, this is the first study to shed light on the multifaceted influences shaping DMI among Chinese patients with CKD, offering actionable insights for effective interventions. By highlighting PMT constructs and sociodemographic factors influential on patients' intention to engage with dietary management, we pave the way for effective strategies to improve dietary adherence and health outcomes of Chinese patients with CKD. Findings from this study can be used to inform the development of theory-based dietary management support programs and ultimately improve the overall health outcomes for patients with CKD.

## Supporting information

**S1 Appendix. STROBE checklist.**
(DOCX)

**S2 Appendix. Global inclusivity checklist.**
(DOCX)

**S3 Appendix. Study questionnaire.**
(DOCX)

**S4 Appendix. Minimal dataset.**
(XLSX)

## Acknowledgments

Special thanks to Di Zhang and Wenshu Ge for administrative support in data collection. We thank all the participants for their assistance in data collection.

## Author contributions

**Conceptualization:** Yueyi Deng, Yitong Huang.

**Data curation:** Huijie Li.

**Formal analysis:** Huijie Li.

**Funding acquisition:** Yueyi Deng.

**Investigation:** Huijie Li, Holly Blake.

**Methodology:** Huijie Li.

**Project administration:** Yitong Huang.

**Resources:** Yueyi Deng.

**Supervision:** Yitong Huang.

**Writing – original draft:** Huijie Li.

**Writing – review & editing:** Yitong Huang, Holly Blake.

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
