## [Decision Letter · Decision Letter 0]

30 Dec 2024

PONE-D-24-37218Predicting Dietary Management Intention of Patients with Chronic Kidney Disease Using Protection Motivation TheoryPLOS ONE

Dear Dr. Huang,

Thank you for submitting your manuscript to PLOS ONE. After careful consideration, we feel that it has merit but does not fully meet PLOS ONE’s publication criteria as it currently stands. Therefore, we invite you to submit a revised version of the manuscript that addresses the points raised during the review process.

We look forward to receiving your revised manuscript.

Kind regards,

Prathap kumar Simhadri, MD

Academic Editor

PLOS ONE

Additional Editor Comments:

Comments from the Editor-

Thank you for submitting the interesting manuscript, and our decision at this time is to " revise the manuscript". Please address all the questions raised by the reviewers and resubmit the revised manuscript.

Please clarify the parameters of the variables mentioned in the Table 1. (Weight in pounds or kilograms, Height in inches, please provide the dollar equivalent of the currency mentioned etc.)

Reviewer 1.

The manuscript is very interesting, it assesses a fundamental aspect of the treatment of patients with CKD.

In my opinion, it would be more intuitive for the reader to insert a table with the items evaluated by the questionnaire, rather than to report them in the text (line 119-128).

Reviewer 2.

I would have like to see sub analysis based on different stages of CKD as dietary restrictions are more restrictive in advanced ckd compared to CKD stage III A patients

It was done only in hospitalized patients and so there is selection bias and would be benefecial to see this hypothesis done in office patients

There were almost 88 % who were married and so unsure if results could be generalized, While the hypotheses are well-stated, presenting the results in a table specifically dedicated to hypothesis testing (e.g., whether each hypothesis is supported or rejected) would add clarity for readers.

Reviewer 3.

If the patient population had any other factors like hypertension, obesity or other cardiovascular risk factors that could make the results skewed. Otherwise the article appears to be interesting idea.

Reviewer 4.

The study is written in fluent English and well articulated. It is written with good flow in almost all the sections.

Methods used were relevant and well conducted.

Statistical analysis - did the study account for CKD stage which can inherently influence the DMI bringing in bias?

weakness remains the cross-sectional nature of the study which was also mentioned in the study limitations.

Overall the study offered valuable information for actionable items and also paved way for further longitudinal research to confirm as well as elaborate on the behavioral aspects behind dietary management intention.

Reviewer 5.

1. Very interesting and tailored approach to enhance adherence to dietary recommendations in Chronic Kidney Disease

2. Also, I have found no similar articles

3. Small word correction on page 9, line 205, 'report' instead of 'repot'

4. I am a proponent of diet modification in CKD and agree that all the current recommendations are generalized to the Western population, especially where access to processed food it very high, which plays a significant role in worsening of several chronic diseases including Diabetes, Hypertension, Heart disease and CKD, and which are more than often interconnected.

5. I feel this study lays the foundation for future large prospective or Randomized Control Trials on this topic and encourage more researchers from different continents to use similar or different approaches to study the factors influencing dietary habits of different ethnicities so that a targeted approach can be employed to reduce the global burden of CKD and ESKD.

Reviewer 6.

1) Keep the Dietary Management Intentions- consistent

Not Intention in one place and intentions in other places

2) Chronic kidney disease (CKD) is a pathological condition in which a gradual loss

of kidney function occurs over five stages classified by estimated glomerular filtration

rate . This is a clinical classification and classified into 5 stages- needs correction

3) As patients in early stages of CKD might have few signs or symptoms, they

may feel reluctant to seek and follow medical advice on managing the condition.

This statement makes more sense if it says less instead of few.

Also, this study seems more appropriate on an outpatient basis. Any explanation as to why this was intended to be done as inpatient

Reviewers' comments:

Reviewer's Responses to Questions

**Comments to the Author**

1. Is the manuscript technically sound, and do the data support the conclusions?

Reviewer #1: Yes

Reviewer #2: Yes

Reviewer #3: Yes

Reviewer #4: Yes

Reviewer #5: Yes

Reviewer #6: Yes

2. Has the statistical analysis been performed appropriately and rigorously? 

Reviewer #1: Yes

Reviewer #2: Yes

Reviewer #3: Yes

Reviewer #4: Yes

Reviewer #5: Yes

Reviewer #6: Yes

3. Have the authors made all data underlying the findings in their manuscript fully available?

Reviewer #1: Yes

Reviewer #2: Yes

Reviewer #3: Yes

Reviewer #4: Yes

Reviewer #5: Yes

Reviewer #6: Yes

4. Is the manuscript presented in an intelligible fashion and written in standard English?

Reviewer #1: Yes

Reviewer #2: Yes

Reviewer #3: Yes

Reviewer #4: Yes

Reviewer #5: Yes

Reviewer #6: Yes

5. Review Comments to the Author

Reviewer #1: The manuscript is very interesting, it assesses a fundamental aspect of the treatment of patients with CKD.

In my opinion, it would be more intuitive for the reader to insert a table with the items evaluated by the questionnaire, rather than to report them in the text (line 119-128).

Reviewer #2: I would have like to see sub analysis based on different stages of CKD as dietary restrictions are more restrictive in advanced ckd compared to CKD stage III A patients

It was done only in hospitalized patients and so there is selection bias and would be benefecial to see this hypothesis done in office patients

There were almost 88 % who were married and so unsure if results could be generalizedWhile the hypotheses are well-stated, presenting the results in a table specifically dedicated to hypothesis testing (e.g., whether each hypothesis is supported or rejected) would add clarity for readers.

Reviewer #3: If the patient population had any other factors like hypertension, obesity or other cardiovascular risk factors that could make the results skewed. Otherwise the article appears to be interesting idea.

Reviewer #4: The study is written in fluent English and well articulated. It is written with good flow in almost all the sections.

Methods used were relevant and well conducted.

Statistical analysis - did the study account for CKD stage which can inherently influence the DMI bringing in bias?

weakness remains the cross-sectional nature of the study which was also mentioned in the study limitations.

Overall the study offered valuable information for actionable items and also paved way for further longitudinal research to confirm as well as elaborate on the behavioral aspects behind dietary management intention.

Reviewer #5: 1. Very interesting and tailored approach to enhance adherence to dietary recommendations in Chronic Kidney Disease

2. Also, I have found no similar articles

3. Small word correction on page 9, line 205, 'report' instead of 'repot'

4. I am a proponent of diet modification in CKD and agree that all the current recommendations are generalized to the Western population, especially where access to processed food it very high, which plays a significant role in worsening of several chronic diseases including Diabetes, Hypertension, Heart disease and CKD, and which are more than often interconnected.

5. I feel this study lays the foundation for future large prospective or Randomized Control Trials on this topic and encourage more researchers from different continents to use similar or different approaches to study the factors influencing dietary habits of different ethnicities so that a targeted approach can be employed to reduce the global burden of CKD and ESKD.

Reviewer #6: 1) Keep the Dietary Management Intentions- consistent

Not Intention in one place and intentions in other places

2) Chronic kidney disease (CKD) is a pathological condition in which a gradual loss

of kidney function occurs over five stages classified by estimated glomerular filtration

rate . This is a clinical classification and classified into 5 stages- needs correction

3) As patients in early stages of CKD might have few signs or symptoms, they

may feel reluctant to seek and follow medical advice on managing the condition.

This statement makes more sense if it says less instead of few.

Also, this study seems more appropriate on an outpatient basis. Any explanation as to why this was intended to be done as inpatient

6. PLOS authors have the option to publish the peer review history of their article (what does this mean? ). If published, this will include your full peer review and any attached files.

**Do you want your identity to be public for this peer review?** For information about this choice, including consent withdrawal, please see our Privacy Policy .

Reviewer #1: No

Reviewer #2: No

Reviewer #3: No

Reviewer #4: **Yes: ** Babu Sriram Maringanti

Reviewer #5: **Yes: ** Vikas Vujjini

Reviewer #6: **Yes: ** Hari Naga Garapati

---

## [Author Response · Author response to Decision Letter 1]

16 Jan 2025

Reviewer 1

The manuscript is very interesting, it assesses a fundamental aspect of the treatment of patients with CKD.

In my opinion, it would be more intuitive for the reader to insert a table with the items evaluated by the questionnaire, rather than to report them in the text (line 119-128).

RESPONSE:

Thank you for the valuable suggestion. Reporting the items in the main text also allows us to cite the sources of scales for different constructs and explain any adaptions made. Considering the length of the manuscript, we have included the full table of survey items/variables in S3 Appendix. This placement ensures ease of reading for the manuscript while facilitating accessibility for researchers who may wish to replicate the study or reuse the study material in the future.

Reviewer 2.

I would have like to see sub analysis based on different stages of CKD as dietary restrictions are more restrictive in advanced ckd compared to CKD stage III A patients

It was done only in hospitalized patients and so there is selection bias and would be benefecial to see this hypothesis done in office patients

There were almost 88 % who were married and so unsure if results could be generalized, While the hypotheses are well-stated, presenting the results in a table specifically dedicated to hypothesis testing (e.g., whether each hypothesis is supported or rejected) would add clarity for readers.

RESPONSE:

Thank you very much for your thoughtful feedback.

1. CKD Stages and Sub-analyses:

We have additionally included CKD stage as a covariate in our regression model and confirmed that it has a significantly positive association with dietary management intention (DMI). A discussion on this finding is also added (lines 333-340). However, as there were <100 respondents in some CKD stages, sub-group analysis was not feasible. We did perform exploratory analysis to see if there was any significant CDK stage × PMT construct interaction, but found none.

2. Selection bias and generalizability to outpatients:

Thank you for your comment. We have acknowledged this as a limitation and explained reasons for choosing the inpatient setting (Lines 359-367).

3. Marital Composition:

Regarding the sample's marital composition, we believe it aligns well with China's broader adult population. According to the Seventh National Census results announced in 2021, 220 million people aged 20 or above were unmarried, accounting for 13.3% of this age group. Therefore, 86.7% of adults in China were married, a proportion similar to the 88% married in our study sample (Source: https://www.chinadaily.com.cn/a/202202/14/WS62099b7fa310cdd39bc8652d.html).

4. Hypothesis Testing Table:

Lastly, we appreciate your suggestion to present the results in a dedicated table specifically for hypothesis testing (Table5). We have incorporated this into the manuscript for enhanced clarity and to facilitate readers' understanding of the results.

Reviewer 3.

If the patient population had any other factors like hypertension, obesity or other cardiovascular risk factors that could make the results skewed. Otherwise the article appears to be interesting idea.

RESPONSE:

Thank you for raising this important point. CKD dietary management is widely recognized as a complex issue, partly due to the commonly observed comorbidities (e.g. diabetes, hypertension), each of which may have its own dietary restrictions. We have highlighted this in our manuscript, both in the introduction (Lines 60-63) and discussion (Line 310-313). While these comorbidities do affect dietary management complexity and thus potentially intention, they reflect real-world patient populations rather than inducing skewness. Indeed, one of the dietary approaches recommended for CKD patients in practice is the DASH (Dietary Approaches to Stop Hypertension). Just as Reviewer 5 has mentioned, chronic diseases such as diabetes, hypertension, heart disease, and CKD are often interconnected and influenced by dietary behaviors, which makes our study particularly timely and interesting.

Reviewer 4.

The study is written in fluent English and well articulated. It is written with good flow in almost all the sections. Methods used were relevant and well conducted. Statistical analysis - did the study account for CKD stage which can inherently influence the DMI bringing in bias? weakness remains the cross-sectional nature of the study which was also mentioned in the study limitations. Overall the study offered valuable information for actionable items and also paved way for further longitudinal research to confirm as well as elaborate on the behavioral aspects behind dietary management intention.

RESPONSE:

Thank you for your positive and constructive feedback. In response to your suggested change, we have added CKD stage as a covariate to our regression model and confirmed that it has a significantly positive association with dietary management intention (DMI). This result is indeed interesting and discussed in detail in the manuscript (lines 333-340). Thank you again.

Reviewer 5.

1. Very interesting and tailored approach to enhance adherence to dietary recommendations in Chronic Kidney Disease

RESPONSE: Thank you for your valuable time and positive feedback.

2. Also, I have found no similar articles

RESPONSE: Thank you again.

3. Small word correction on page 9, line 205, 'report' instead of 'repot'

RESPONSE: Many thanks for catching this typo. We have made corrections accordingly.

4. I am a proponent of diet modification in CKD and agree that all the current recommendations are generalized to the Western population, especially where access to processed food it very high, which plays a significant role in worsening of several chronic diseases including Diabetes, Hypertension, Heart disease and CKD, and which are more than often interconnected.

RESPONSE: Many thanks for your support and for sharing your valuable perspective.

5. I feel this study lays the foundation for future large prospective or Randomized Control Trials on this topic and encourage more researchers from different continents to use similar or different approaches to study the factors influencing dietary habits of different ethnicities so that a targeted approach can be employed to reduce the global burden of CKD and ESKD.

RESPONSE: Many thanks for your encouraging remarks and support. We fully agree and are excited to share our findings with the broader research community as soon as possible and hope it inspires further exploration in this area.

Reviewer 6.

1) Keep the Dietary Management Intentions- consistent

Not Intention in one place and intentions in other places

RESPONSE: Thank you for pointing this out. We have ensured consistency by using "Dietary Management Intention" throughout the manuscript.

2) Chronic kidney disease (CKD) is a pathological condition in which a gradual loss of kidney function occurs over five stages classified by estimated glomerular filtration rate . This is a clinical classification and classified into 5 stages- needs correction

RESPONSE: Thank you for highlighting this. We have revised the sentence for clarity and accuracy as follows “Chronic kidney disease (CKD) is a pathological condition in which a gradual loss of kidney function occurs. It is clinically classified into five stages based on estimated glomerular filtration rate (eGFR).

3) As patients in early stages of CKD might have few signs or symptoms, they may feel reluctant to seek and follow medical advice on managing the condition. This statement makes more sense if it says less instead of few.

RESPONSE: Many thanks for the careful observation. However, "few" seems to suit better here because it properly applies to countable nouns like "signs or symptoms."

Also, this study seems more appropriate on an outpatient basis. Any explanation as to why this was intended to be done as inpatient

RESPONSE: Thank you for your question, which is extremely valid and appreciated. Given the literacy levels of our study respondents, it was preferrable they were given sufficient time to read and understand the survey questions and ask the researcher for explanations if needed. Outpatient settings in China are often very crowded and hectic, which would inevitably hinder the effectiveness of data collection. That is the primary reason for choosing the inpatient setting. Moreover, it is common for CKD patients to alternate between inpatient and outpatient care, especially given that the hospital was a Chinese medicine hospital specialized in chronic disease management. Patients are discharged to community care once their condition improves following inpatient treatment; but they may require rehospitalization if their adherence to self-management is poor. This motivated us to do this study in the first place.

We have added a brief clarification to Lines 359-367. Many thanks again, for raising this question.

---

## [Editor Report · Decision Letter 1]

22 Jan 2025

PONE-D-24-37218R1Predicting dietary management intention of patients with chronic kidney disease using protection motivation theoryPLOS ONE

Dear Dr. Huang,

Thank you for submitting your manuscript to PLOS ONE. After careful consideration, we feel that it has merit but does not fully meet PLOS ONE’s publication criteria as it currently stands. Therefore, we invite you to submit a revised version of the manuscript that addresses the points raised during the review process.

**Thank you for revising the manuscript and it needs further minor revision before it can be accepted. Please verify the parameters of the variables mentioned in the parenthesis for the Table 1 and resubmit the revised manuscript. **

We look forward to receiving your revised manuscript.

Kind regards,

Prathap kumar Simhadri, MD

Academic Editor

PLOS ONE

**Journal Requirements:**

**Additional Editor Comments:**

Thank you for revising the manuscript and addressing all the questions raised by the reviewers. The parameters mentioned in the parenthesis for the variables Height and Weight need to be revised. Please revise the manuscript further before it could be accepted.

---

## [Author Response · Author response to Decision Letter 2]

25 Jan 2025

Dear Editor and Reviewer,

Many thanks again for your valuable time and attention to detail, which we really appreciate.

Best Regards,

---

## [Editor Report · Decision Letter 2]

18 Feb 2025

Predicting dietary management intention of patients with chronic kidney disease using protection motivation theory

PONE-D-24-37218R2

Dear Dr. Huang,

We’re pleased to inform you that your manuscript has been judged scientifically suitable for publication and will be formally accepted for publication once it meets all outstanding technical requirements.

Kind regards,

Prathap kumar Simhadri, MD

Academic Editor

PLOS ONE
---

## [Editor Report · Acceptance letter]

PONE-D-24-37218R2

PLOS ONE

Dear Dr. Huang,

I'm pleased to inform you that your manuscript has been deemed suitable for publication in PLOS ONE. Congratulations! Your manuscript is now being handed over to our production team.

Kind regards,

on behalf of

Dr. Prathap kumar Simhadri

Academic Editor

PLOS ONE